# Determinants of knowledge, attitude, and practice among patients with type 2 diabetes mellitus: A cross-sectional multicenter study in Tanzania

**Angelina A. Joho** [1]*, **Frank Sandi** [2], **James J. Yahaya** [3]

1 Department of Clinical Nursing, School of Nursing and Public Health, The University of Dodoma, Dodoma, Tanzania, 2 Department of Ophthalmology, School of Medicine and Dentistry, The University of Dodoma, Dodoma, Tanzania, 3 Department of Pathology, School of Health Sciences, Soroti University, Soroti, Uganda

* johoangeljoho@yahoo.co.uk

## Abstract

Improvement of primary care for patients with type 2 diabetes mellitus (T2DM) through the promotion of good knowledge, attitude, and practice is of paramount importance for preventing its related complications. This study aimed to assess the levels of knowledge, attitude, and practice and associated factors among patients with T2DM. This was a cross-sectional multicenter hospital-based study that included 979 patients from 8 health facilities in Tanzania. A standardized semi-structured interviewer-administered questionnaire was used to extract the required data. Factor analysis was used to determine the level of knowledge, attitude, and practice. Multivariable analysis under binary logistic regression analysis was used to determine the predictors of knowledge, attitude, and practice. P<0.05 was considered significant. The levels of adequate knowledge, positive attitude, and appropriate practice were 62.1%, 54%, and 30.9%, respectively. Being self-employed (AOR = 1.74, 95% CI = 0.28–0.91, p = 0.040) predicted adequate knowledge. Being male (AOR = 1.46, 95% CI = 1.06–2.01, p = 0.021 and visiting regional hospitals (AOR = 2.17, 95% CI = 1.33–2.51, p = 0.013) were predictors of positive attitude. Residing in rural areas and not having adequate knowledge of diabetes were less likely associated with appropriate practice. This study has shown a significantly low level of appropriate practice among patients with T2DM towards general issues on diabetes, risk factors, and related complications. Therefore, emphasis should be placed on improving good practices that can help prevent related complications.

## Introduction

Diabetes mellitus (DM) is a non-communicable disease (NCD) that remains a major threat and a health problem of public concern globally. Previously, DM was not found to be a health problem of public concern in developing countries like Tanzania. However, from 2015, the prevalence of DM was reported to be extremely raising [1]. According to the World Bank report, the prevalence of DM in Tanzania was reported to be 12.3% in 2021, in individuals with age between 20 and 79 years [2]. The International Diabetes Federation (IDF) reported

**Data Availability Statement:** The dataset used for this study is restricted by the Research Ethical Committee of the institution detail due to containing sensitive patient information, however,

it can be accessed upon reasonable request from the Directorate of Research Publication, and Consultancy (DRPC), University of Dodoma, P. O. Box 259, Dodoma, Tanzania. drpc@udom.ac.tz.

**Funding:** The authors received no specific funding for this work.

**Competing interests:** The authors have declared that no competing interests exist.

that by 2025 the number of individuals with DM in Tanzania is expected to increase by 3.7% [3]. This tremendous and alarming drastic increase in the incidence of DM in developing countries in which Tanzania is included has been linked to substantial demographic change from traditional ways of living to westernized and urbanized [4].

The prevalence of adequate knowledge in developing countries in which the prevalence of DM has been shown to be raising is quite lower than that in developed countries [5–7]. For example, in Bangladesh, the level of good knowledge among diabetics was 15% which is extremely lower than 92.9% of good knowledge among diabetics which was reported in Slovenia [1, 8]. Also, the attitude has a direct impact on prevention of the complications related to T2DM. This is due to the fact that, negative attitude contributes to failure of medication adherence, unhealthy lifestyle which all together have a direct causal-effect relationship with T2DM complications [9, 10]. Diabetics with positive attitude towards T2DM related complications are more likely to confer with preventive measures regarding T2DM complications [1, 11].

Concerning practice towards prevention of T2DM complications; studies have shown that appropriate practice towards prevention of T2DM among diabetics helps to decipher occurrence of morbidity and mortality. It has been shown that, practice is negatively affected by both inadequate knowledge and negative attitude [12].

In Tanzania, there is a scarcity of multicenter studies that address knowledge, attitude, and practice among patients with T2DM. Therefore, this creates a gap in knowledge, attitude and practice for a population that involves study participants taken from multiple study sites within the country which has a high chance of giving representative information. We aimed to explore the levels of knowledge, attitude and practice towards risk factors, complications, and preventive measures of T2DM among diabetics in Tanzania.

## Materials and methods

### Study design and study area

This was a cross-sectional analytical hospital-based study which involved multiple centers in Tanzania. The study was conducted at 8 study sites which included three district hospitals (Bagamoyo, Nzega and Haydom) and five regional referral hospitals (Iringa, Dodoma, Shinyanga, Tabora and, Mawenzi). Out of the 8 health facilities, only one health facility (Haydom district hospital) was the only faith-based health facility. The visit to diabetes care clinics at the selected study sites is usually done based on the appointment given by the physicians to the patients. Patients usually are evaluated for medication adherence, glycemic control and possible T2DM related complications for every visit.

### Study participants

In this study, we included patients with T2DM. The criteria for recruiting the participants included patients aged 18 years and above, having a duration of at least 1 year since diagnosis, and those who agreed to sign written informed consent. We excluded all patients who denied to sign written informed consent, patients with gestational diabetes mellitus, patients younger than 18 years, and all patients who reported to have been involved or participated in sessions for raising of awareness and knowledge on diabetes were excluded in order to understand the kind of efforts that the Ministry of Health has to undertake in order to sensitize the vulnerable population.

### Sampling method

The sample size was determined using a formula for calculating prevalence for a single population in a cross-sectional study design which was developed by Leslie Kish: $n = z^2 p (1-p)/\varepsilon^2$

[13] where: n = sample size, z = standard normal deviate (1.96) on using 95% CI, p = expected prevalence of the outcome which was assumed to be 53% for knowledge from a previous study [12]. Considering a contingency of 20%, for non-respondents, the total sample size was 979 of patients with T2DM. Convenience sampling method was used to recruit the study participants whereby study participants were consecutively recruited until the required total sample size was complete.

## Data collection method and tools

Data collection was done by the authors and 8 research assistants who were registered nurses. Data were collected after the participants had signed informed consent. The process of data collection was carried out in a secluded room for maintaining privacy of study participants. We adapted and modified questionnaires from the previous studies which were done in Ethiopia [11] for assessment of attitude and practice and Malaysia [14] for assessment of level of knowledge (additional file 1). This was followed by modifying some items from the questionnaires including simplifying a number of words to make the questionnaire understandable. Then we translated the questionnaire into Swahili local language. The questionnaire was validated by calculating Cronbach's alpha whose value was 8.74 after we had conducted a pilot study using 20 participants from a health facility different from the study sites.

## Measurement of variables

**Knowledge.** A total of 25 items were used to measure the level of knowledge. The items had two responses "Yes/No" and each response was scored 1 point. In order to check for internal consistency of validity of the tool, we performed factor analysis statistic so as to reduce the weak items. The Kaiser-Mayer-Olkin (KMO) measure of sampling adequacy was 0.878, Bartlet's test of sphericity was 3679.530, and p-value <0.001. The total variance explained (initial Eigenvalues) had seven components with cumulative percentage of 68.277. Knowledge scores were approximately normally distributed with Shapiro-Wilk test value of statistic of 0.503 and p-value <0.001. The mean score was considered to be the cut-off point for those with adequate and inadequate knowledge. The mean score was 10.35 ± 2.14 (range: 0–25), therefore, participants with less than mean score were considered to have inadequate knowledge and vice versa.

**Attitude.** Attitude was assessed using 9 items which were measured on five points Likert scale. 1-strongly agree, 2-agree, 3-neither agree nor disagree, 4-disagree, and 5-strongly disagree. Points 1 and 2 implied positive attitude whereas points 3 through 5 implied negative attitude. In checking for internal consistency of validity of the tool, factor analysis statistic was performed so as to reduce the weak items. The Kaiser-Mayer-Olkin (KMO) measure of sampling adequacy was 0.506, Bartlet's test of sphericity was 2003.303, and p-value <0.001. The total variance explained (initial Eigenvalues) had nine components with cumulative percentage of 71.040. Attitude scores were approximately normally distributed with Shapiro-Wilk test value of statistic of 0.958 and p-value <0.001. The cut-off point for positive and negative attitude was a mean score which was 5.497 ± 1.986. All participants with points equal or greater than the mean score were considered to have positive attitude and vice versa.

**Practice.** Practice was assesses using 10 items with responses "Yes/No" and each response was given 1 point. Determination of internal consistency of validity of the tool was performed after running factor analysis statistics to reduce all weak items. KMO measure of sampling adequacy was 0.500, Bartlet's test of sphericity was 1056.463, and p-value <0.001. The total variance explained (initial Eigenvalues) had nine components with cumulative percentage of 79.135. Practice scores were approximately normally distributed with Shapiro-Wilk test value of statistic of 0.951 and p-value <0.001. The cut-off point for considering appropriate and

inappropriate practice in this study was the mean score (4.72 ± 1.418) with range of 0–10 points. Participants with less than the mean score points were termed to have inappropriate practice and vice versa.

## Data analysis

The collected data were analyzed using SPSS version 23.0. For descriptive statistic, frequency and percentages were used to summarize categorical variables and continuous variables were summarized in mean ± standard deviation (SD). Factor analysis was used to determine the levels of knowledge, attitude, and practice. Multivariable analysis under binary logistic regression analysis was used to determine the predictors for knowledge, attitude, and practice by calculating the adjusted odds ratios (AORs) at 95% confidence interval (CI). A two-tailed $p < 0.05$ was considered statistically significant.

## Ethical approval

We obtained ethical approval from the Institutional research committee of the University of Dodoma. Additionally, we obtain permission for data collection from the medical officer in-charges (MOIs).

## Results

### Sociodemographic characteristics of participants

The sociodemographic characteristics of the study participants are shown in Table 1. A total of 979 study participants were included in the present study with mean age of 51.1 ± 15.4 years. Over half 520 (53.2%) had 51 years and above and also over half of the participants 517 (52.8%) were women. More than half 500 (51.1%) of the study participants had attained primary education level. Majority of participants 596 (60.9%) were residing in rural area. Of all the study participants, 243 (24.8%) had various types of comorbidities. Also, majority 631 (64.5%) of the study participants had ≥60 months since when they were diagnosed with mean of 60.6 ± 57.2 in months.

### Levels of knowledge, positive attitude and appropriate practice

Fig 1 presents the levels of adequate knowledge, positive attitude, and appropriate practice. Of the three domains studied, knowledge was the domain with the highest proportion in whom majority (n = 608, 62.1%, 95% CI = 47.47–48.05) of the participants had adequate knowledge followed by positive attitude which was found in (n = 529, 54.0%, 95% CI = 19.15–19.51) of the participants. Appropriate practice was reported in only one-third (n = 303, 30.9%, 95% CI = 16.09–16.26) of the study participants.

### Factors associated with knowledge among study participants

In the multivariable logistic analysis in Table 2, self-employed patients were 1.74 (95% CI = 1.12–2.70, p = 0.040) times more likely to have adequate diabetes knowledge than unemployed or government-employed patients. Similarly, patients who were visiting district hospitals for their treatment were 35% significantly less likely to have appropriate knowledge on diabetes compared with patients who were visiting referral hospitals for their treatment (AOR = 0.65, 95% CI = 0.44–0.98, p = 0.037).

**Table 1.  Sociodemographic characteristics of participants (N = 979).**

| Variable | n (%) |
|---|---|
| **Age (years)** | |
| 18–30 | 113 (11.5) |
| 31–50 | 346 (35.3) |
| ≥51 | 520 (53.2) |
| **Sex** | |
| Men | 462 (47.2) |
| Women | 517 (52.8) |
| **Level of education** | |
| Informal | 194 (19.8) |
| Primary | 500 (51.1) |
| Secondary | 217 (22.2) |
| Tertiary | 68 (6.9) |
| **Occupation** | |
| Unemployed | 119 (12.1) |
| Self-employed | 649 (66.3) |
| Employed | 211 (21.6) |
| **Marital status** | |
| Single | 155 (15.8) |
| Married/cohabiting | 656 (67.0) |
| Divorced/separated | 68 (6.9) |
| Widow/widower | 100 (10.2) |
| **Residence** | |
| Rural | 596 (60.9) |
| Urban | 383 (39.1) |
| **Health facility ownership** | |
| Government | 818 (83.6) |
| Private | 161 (16.4) |
| **Level of health facility** | |
| District | 145 (14.8) |
| Regional | 287 (29.3) |
| Referral | 547 (55.9) |
| **Comorbidities** | |
| Yes | 243 (24.8) |
| No | 736 (75.2) |
| **Duration of illness (months)** | |
| ≤ 60 | 631 (64.5) |
| > 60 | 348 (35.5) |

## Factors associated with attitude among study participants

Also, in the multivariable logistic analysis in Table 3, for factors associated with attitude levels towards diabetes we found that, male patients were 1.46 times more likely to have positive attitude than female patients (AOR = 1.46, 95% CI = 1.06–2.01, p = 0.021). Patients who were visiting regional hospitals for their treatment were 2.17 times more likely to have positive attitude than those who were visiting district or referral hospitals for their treatment (AOR = 2.17, 95% CI = 1.33–2.51, p = 0.013). Furthermore, it was found that, young patients aged between 18 and 30 years, those with no formal education, and patients with inadequate knowledge towards diabetes were 49% (AOR = 0.51, 95% CI = 0.28–0.91, p = 0.023), 64% (AOR = 0.36, 95%

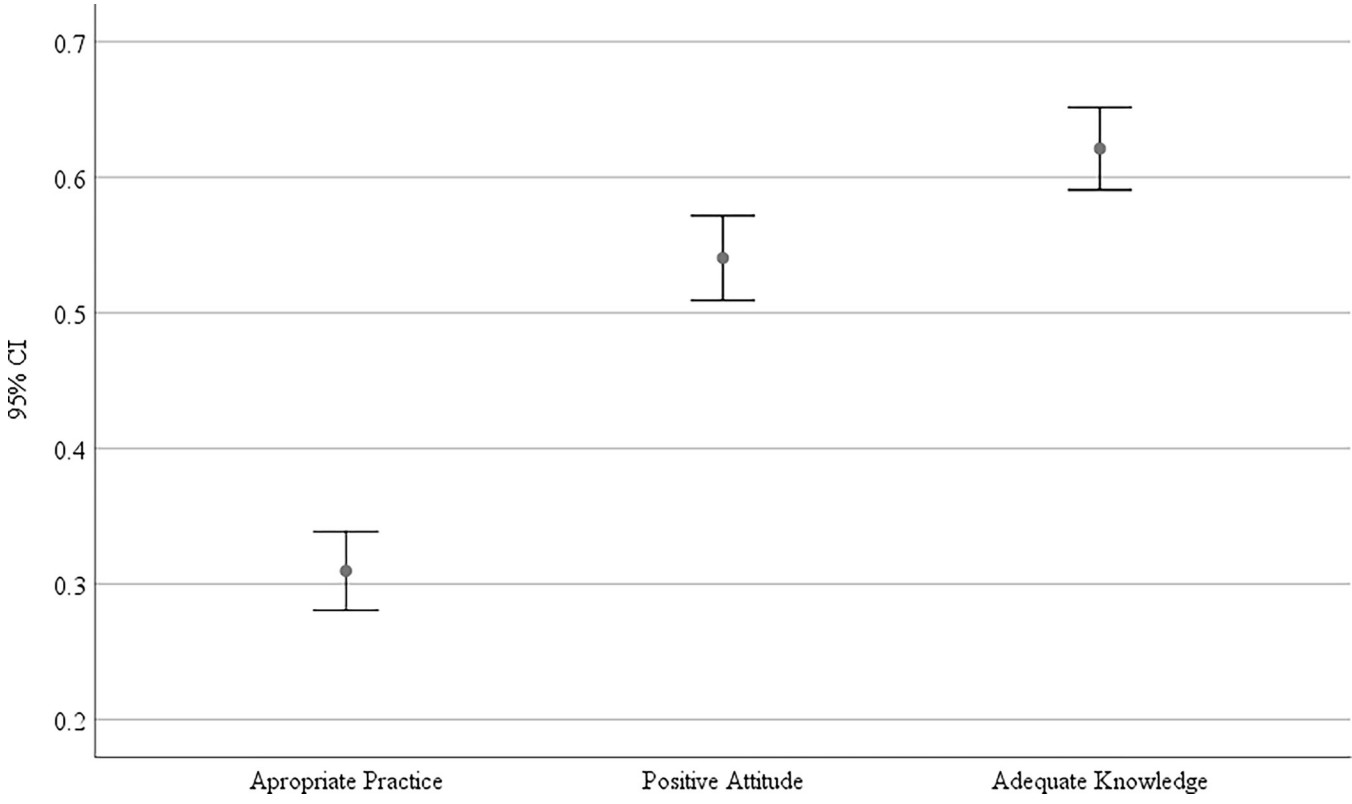

**Fig 1. The levels of adequate knowledge, positive attitude, and appropriate practice.**

CI = 0.18–0.73, p = 0.005), and 68% (AOR = 0.32, 95% CI = 0.24–0.42, p<0.001) less likely to have positive attitude towards diabetes compared with old patients, patients with formal education, and those with adequate knowledge, respectively (Table 3).

### Factors associated with practice among study participants

Regarding factors associated with practice of patients towards diabetes, it was found that residing in rural areas and having inadequate knowledge on diabetes were associated with 47% (AOR = 0.53, 95% CI = 0.38–0.72, p<0.001) and 10% (AOR = 0.90, 95% CI = 0.06–0.14, p<0.001) less chance of having appropriate practice towards diabetes, respectively. Patients who were visiting regional hospitals for their treatment were 50% (AOR = 0.50, 95% CI = 0.34–0.72, p<0.001) less likely to have appropriate practice towards diabetes compared with patients who were visiting referral hospitals (Table 4).

### Discussion

Patients with T2DM have increased risk of developing a number of complications which may either be early or late complications. Such complications are more common in developing countries where the levels of knowledge, attitude and practices have been found to be low. Increased level of appropriate knowledge, good attitude, and appropriate practices among diabetics help to improve the life of the patients through prevention of development of related complications which usually tend to affect the quality of life of the diabetics. The key findings include a significant high level of appropriate knowledge, moderate good attitude, and markedly low level of appropriate practices regarding diabetes.

**Table 2. Multivariable analysis of factors associated with knowledge towards diabetes mellitus.**

| Variables | Univariate analysis | | Multivariable analysis | |
|---|---|---|---|---|
| | UOR (95% CI) | P | AOR (95% CI) | p |
| **Age (years)** | | | | |
| 18–30 | 2.05 (1.30–3.21) | 0.002 | 1.64 (0.90–2.99) | 0.109 |
| 31–50 | 1.39 (1.05–1.85) | 0.021 | 1.31 (0.96–1.80) | 0.091 |
| ≥51 | 1.00 | | 1.00 | |
| **Sex** | | | | |
| Male | 0.95 (0.73–1.23) | 0.7 | 0.91 (0.70–1.19) | 0.498 |
| Female | 1.00 | | 1.00 | |
| **Level of education** | | | | |
| No informal education | 1.18 (0.66–2.10) | 0.572 | 1.50 (0.76–2.95) | 0.243 |
| Primary | 0.80 (0.48–1.36) | 0.411 | 0.91 (0.49–1.68) | 0.755 |
| Secondary | 1.17 (0.66–2.07) | 0.586 | 1.22 (0.65–2.30) | 0.539 |
| Tertiary | 1.00 | | 1.00 | |
| **Marital status** | | | | |
| Married/cohabiting | 2.22 (1.31–3.76) | 0.003 | 1.60 (0.81–3.11) | 0.175 |
| Divorced/separated | 1.32 (0.87–2.02) | 0.196 | 1.05 (0.65–1.69) | 0.840 |
| Widow/widower | 1.49 (0.78–2.75) | 0.235 | 1.11 (0.57–2.15) | 0.754 |
| Single | 1.00 | | 1.00 | |
| **Occupation** | | | | |
| Self-employed | 0.49 (0.72–1.37) | 0.971 | 1.74 (1.12–2.70) | 0.040 |
| Unemployed | 0.70 (0.73–1.04) | 0.078 | 0.69 (0.99–2.88) | 0.054 |
| Employed | 1.00 | | 1.00 | |
| **Place of residence** | | | | |
| Rural | 0.89 (0.69–1.17) | 0.407 | 1.13 (0.86–1.49) | 0.392 |
| Urban | 1.00 | | 1.00 | |
| **Level of health facility** | | | | |
| District | 0.79 (0.55–1.14) | 0.209 | 0.65 (0.44–0.98) | 0.037 |
| Regional | 1.22 (0.91–1.65) | 0.192 | 1.07 (0.78–1.47) | 0.682 |
| Referral | 1.00 | | 1.00 | |
| **Comorbidities** | | | | |
| Yes | 0.78 (0.58–1.05) | 0.096 | 1.06 (0.76–1.47) | 0.746 |
| No | 1.00 | | 1.00 | |
| **Disease duration (years)** | | | | |
| <5 | 1.13 (0.74–1.72) | 0.567 | 0.86 (0.53–1.37) | 0.746 |
| 5–10 | 0.86 (0.55–1.34) | 0.514 | 0.78 (0.49–1.24) | 0.299 |
| >10 | 1.00 | | 1.00 | |

The level of adequate knowledge on diabetes in this study was higher than 47%, 54.6%, 54%, 43.5%, and 52% which was reported in Benin [12], Uganda [15], India [16], South India [17], and United States of America [18], respectively. However, higher levels of knowledge among diabetics of 72.9% and 75.6% have been reported in Oman and Saudi Arabia, respectively [19, 20]. The discrepancy in level of knowledge observed may be due to difference in the ways of providing health education from different settings globally. Another reason is the difference in methodology used in assessing the level of knowledge among diabetics could also contribute to the difference of the level of knowledge on diabetes. For example, the use of convenience sampling method when recruiting study participants usually leads to selection bias which may influence the results for a particular assessed domain [21]. Also, lack of public

**Table 3. Multivariable analysis of factors associated with attitude towards diabetes mellitus.**

| Variables | Univariate analysis | | Multivariable analysis | |
|---|---|---|---|---|
| | UOR (95% CI) | p | AOR (95% CI) | p |
| **Age (years)** | | | | |
| 18–30 | 0.86 (0.57–1.29) | 0.465 | 0.51 (0.28–0.91) | 0.023 |
| 31–50 | 1.03 (0.78–1.35) | 0.843 | 0.86 (0.62–1.19) | 0.359 |
| ≥51 | 1.00 | | 1.00 | |
| **Sex** | | | | |
| Male | 0.84 (0.66–1.08) | 0.183 | 1.46 (1.06–2.01) | 0.021 |
| Female | 1.00 | | 1.00 | |
| **Level of education** | | | | |
| No informal education | 0.38 (0.21–0.68) | 0.001 | 0.36 (0.18–0.73) | 0.005 |
| Primary | 0.49 (0.28–0.84) | 0.010 | 0.53 (0.27–1.02) | 0.059 |
| Secondary | 0.52 (0.29–0.93) | 0.027 | 1.49 (0.25–3.97) | 0.139 |
| Tertiary | 1.00 | | 1.00 | |
| **Marital status** | | | | |
| Married/cohabiting | 1.58 (0.95–2.63) | 0.075 | 1.64 (0.84–3.22) | 0.149 |
| Divorced/separated | 1.41 (0.93–2.15) | 0.111 | 1.18 (0.72–1.93) | 0.507 |
| Widow/widower | 1.32 (0.71–2.45) | 0.377 | 1.06 (0.54–2.07) | 0.875 |
| Single | 1.00 | | 1.00 | |
| **Occupation** | | | | |
| Self-employed | 1.13 (0.76–1.67) | 0.552 | 0.88 (0.56–1.38) | 0.576 |
| Unemployed | 1.60 (1.02–2.52) | 0.042 | 0.99 (0.58–1.71) | 0.997 |
| Employed | 1.00 | | 1.00 | |
| **Place of residence** | | | | |
| Rural | 1.32 (1.02–1.71) | 0.035 | 1.28 (0.97–1.69) | 0.087 |
| Urban | 1.00 | | 1.00 | |
| **Level of health facility** | | | | |
| District | 0.72 (0.50-.03) | 0.074 | 0.92 (0.61–1.38) | 0.680 |
| Regional | 1.34 (0.99–1.78) | 0.051 | 2.17 (1.33–2.51) | 0.013 |
| Referral | 1.00 | | 1.00 | |
| **Comorbidities** | | | | |
| Yes | 1.25 (0.94–1.68) | 0.126 | 1.24 (0.88–1.73) | 0.215 |
| No | 1.00 | | 1.00 | |
| **Disease duration (years)** | | | | |
| <5 | 0.92 (0.61–1.39) | 0.693 | 0.92 (0.24–0.42) | 0.734 |
| 5–10 | 0.98 (0.63–1.51) | 0.925 | 1.01 (0.63–1.62) | 0.965 |
| >10 | 1.00 | | 1.00 | |
| **Knowledge** | | | | |
| Inappropriate | 0.34 (0.26–0.44) | <0.001 | 0.32 (0.24–0.42) | <0.001 |
| Appropriate | 1.00 | | 1.00 | |

awareness which is based on self-need and belief regarding risk factors and complications of T2DM has strongly been associated with low level of knowledge among patients with T2DM [22].

In this study, being employed particularly self-employed was associated with appropriate knowledge on diabetes. This is similar to the findings in the studies done elsewhere [12, 14, 23, 24]. For example, in the study of Abbasi et al in Malaysia, it was shown that patients who had government employment or those who were self-employed were associated significantly with

**Table 4. Multivariable analysis of factors associated with practice towards diabetes mellitus.**

| Variables | Univariate analysis | | Multivariable analysis | |
|---|---|---|---|---|
| | UOR (95% CI) | p | AOR (95% CI) | p |
| **Age (years)** | | | | |
| 18–30 | 1.64 (1.09–2.50) | 0.024 | 1.38 (0.73–2.62) | 0.327 |
| 31–50 | 1.38 (1.03–1.85) | 0.033 | 1.25 (0.86–1.80) | 0.237 |
| ≥51 | 1.00 | | 1.00 | |
| **Sex** | | | | |
| Male | 0.89 (0.68–1.17) | 0.407 | 0.94 (0.56–1.23) | 0.350 |
| Female | 1.00 | | 1.00 | |
| **Level of education** | | | | |
| No informal education | 1.13 (0.62–2.06) | 0.697 | 1.22 (0.57–2.62) | 0.602 |
| Primary | 1.05 (0.60–1.83) | 0.868 | 1.38 (0.68–2.80) | 0.374 |
| Secondary | 1.12 (0.62–2.03) | 0.711 | 1.26 (0.61–2.60) | 0.535 |
| Tertiary | 1.00 | | 1.00 | |
| **Marital status** | | | | |
| Married/cohabiting | 1.22 (0.72–2.09) | 0.460 | 0.72 (0.34–1.52) | 0.385 |
| Divorced/separated | 0.95 (0.60–1.50) | 0.820 | 0.83 (0.47–1.46) | 0.512 |
| Widow/widower | 0.99 (0.51–1.94) | 0.987 | 1.01 (0.47–2.19) | 0.975 |
| Single | 1.00 | | 1.00 | |
| **Occupation** | | | | |
| Self-employed | 0.86 (0.57–1.30) | 0.483 | 0.79 (0.47–1.32) | 0.372 |
| Unemployed | 0.90 (0.56–1.50) | 0.663 | 0.92 (0.50–1.70) | 0.784 |
| Employed | 1.00 | | 1.00 | |
| **Place of residence** | | | | |
| Rural | 0.59 (0.44–0.79) | <0.001 | 0.53 (0.38–0.72) | <0.001 |
| Urban | 1.00 | | 1.00 | |
| **Level of health facility** | | | | |
| District | 1.06 (0.72–1.56) | 0.760 | 1.18 (0.74–1.88) | 0.400 |
| Regional | 0.60 (0.43–0.83) | 0.002 | 0.50 (0.34–0.72) | <0.001 |
| Referral | 1.00 | | 1.00 | |
| **Comorbidities** | | | | |
| Yes | 1.01 (0.74–1.38) | 0.973 | 0.83 (0.56–1.21) | 0.351 |
| No | 1.00 | | 1.00 | |
| **Disease duration (years)** | | | | |
| <5 | 1.19 (0.77–1.85) | 0.440 | 1.14 (0.65–1.94) | 0.689 |
| 5–10 | 0.90 (0.56–1.46) | 0.679 | 0.93 (0.53–158) | 0.749 |
| >10 | 1.00 | | 1.00 | |
| **Knowledge** | | | | |
| Inappropriate | 0.11 (0.07–0.16) | <0.001 | 0.90 (0.06–0.14) | <0.001 |
| Appropriate | 1.00 | | 1.00 | |

good knowledge towards diabetes compared with patients who were unemployed [14]. In another study which was done in Pakistan by Gillani et al it was also reported that patients who had high social economic status (SES) were 1.25 times more likely to have good knowledge than patients with low SES [23].

Furthermore, it was observed that patients who were attending district hospitals for their treatment were less likely to have adequate diabetes knowledge compared with patients who were visiting either regional or referral hospitals for their treatment. This may be explained by

the fact that, most of district hospitals are located in rural areas. Studies have shown that patients with T2DM who reside in rural areas are more likely to have poor knowledge towards diabetes than those who reside in urban areas [1, 23]. This is also similar to the findings in the present study in which patients who were residing in rural areas were more likely to have inadequate knowledge towards diabetes than patients who were from urban areas. Considering Tanzania, where majority of people reside in rural areas, there is a high possibility that a large number of individuals in the country could be having low level of knowledge on diabetes. This was reflected by our findings regarding practices on diabetes which was markedly low. Availability of various sources of information regarding T2DM in urban areas including newspapers, radio, television, and access to internet contribute greatly to increased awareness and knowledge on diabetes among patients with T2DM who live in urban areas.

Concerning attitude among participants in this study, half of the patients had positive attitude towards diabetes. This finding is to the findings in the studies done in Benin and Ethiopia [11, 12]. However, other studies have reported higher levels of positive attitude of 65.2%, 67.2%, and 78% among patients with T2DM [1, 11, 25]. Low levels of positive attitude of 20.1% and 27.6% among patients with T2DM have also been reported in other studies [26, 27].

The variation in level of positive or good attitude towards diabetes observed in various studies may be contributed by a number of factors including sociodemographic and behavioral factors [11]. Another reason may be due to the difference in level of knowledge among study participants. For example, in the study done in Bangladesh reported high level of knowledge of 83% compared to 62.1% which was found in the present study; the level of positive attitude in the study done in Bangladesh was higher than the prevalence of positive attitude found in the present study [1]. Other studies also have shown that high level of knowledge among study participants is also associated with positive attitude towards diabetes [1, 12]. Additionally, lifestyle behaviors and cultural factors play a key role in influencing the attitude of patients towards various diseases including DM [26, 27]. In Tanzania, like many other developing countries, there is a challenge to adherence to regular follow-up [28] as well as low medication adherence to anti-diabetes medication [29] due to use of herbals and believes of watch doctors [30].

Regarding predictors of positive attitude towards diabetes in this study, it was found that being male was associated with positive attitude. Abbasi et al [14] and Gillani et al [23] similarly reported that males were associated with good attitude towards diabetes [14]. In another study done in Brazil there were more males with positive attitude than females, however, the difference did not reach statistical significance [31]. However, in the studies of Alaofe et al and Fatema et al it was found that there was no association between sex and attitude of T2DM patients towards diabetes [1, 12].

Furthermore, it was also found that patients who had inadequate knowledge towards diabetes had less odds of having positive attitude compared with patients who had adequate knowledge on diabetes. This is comparable with findings from many studies [12, 24, 32]. Also, it was observed that level of education was associated with positive attitude towards diabetes in which patients who had no formal education had less odds for having positive attitude towards diabetes than odds for patients with formal education at different levels. Similarly, other previous studies have also shown a positive association between increased level of education and positive attitude [1, 11, 12]. However, some studies have shown lack of association between level of education and positive attitude on diabetes [23, 33]. This discrepancy may be due to lack of uniformity in grading or scoring of level of education and patients having various levels of understanding regarding diabetes which brings heterogeneity.

Appropriate or good practice towards diabetes in this study was low. This is similar to the finding in the study done in Bangladesh [1]. However, high levels of appropriate practice

towards diabetes of 52.4%, 60%, and 62.6% were reported in India, Malaysia, and Bangladesh, respectively [34–36]. The variation of the proportion of appropriate practices for the compared studies may be due to the difference in methodology used in scoring appropriate practices. For example, in a study which reported lower level of practice of 16%; assessment of practice included three scores (moderate, good, and poor practice) [37]. This is different from the method that was used to assess appropriate practice in the present study in which assessment of appropriate practice was based on two categories (either appropriate or inappropriate practice). Therefore, lack of uniform approach in assessing good practice is one of the reasons for the difference in levels of good practice towards diabetes among T2DM patients.

Appropriate practice towards diabetes helps to prevent development of related complications through various ways. However, this important domain is usually affected by a number of factors. For instance, in this study, patients with inappropriate knowledge on diabetes practices were less likely to have appropriate practice than patients who had appropriate knowledge on diabetes. This was similar to the finding in previous studies [14, 24, 38]. Living in rural areas was also associated with decreased odds of appropriate practice among patients on diabetes. Residing in rural areas may be linked to low standards of living and decreased ways of getting information on diabetes, low SES, and cultural factors which normally affect practice for patients towards diabetes [39, 40].

## Study limitations

The findings from this study may not be generalizable because the study participants were hospital-based. This may not have been the same if participants would have been obtained from the general population. Also, there was a challenge with recall bias because the participants were required to recall the past information. Additionally, the use of convenience sampling in this study did not provide an equal chance for patients who were available during the study period, hence causing selection bias.

## Conclusion

This study has shown high knowledge, moderate positive attitude, and significantly low level of appropriate practice towards diabetes. Being self-employed and visiting district hospitals were predictors of adequate knowledge. Moreover, being male, visiting regional hospitals, and having adequate knowledge on diabetes were predictors of positive attitude. Residing in rural areas and not having adequate knowledge on diabetes were less likely associated with appropriate practice. Therefore, this study has revealed that, it is important and necessary to raise awareness and knowledge of patients with diabetes for them to have positive attitude as well as appropriate practices.

## Acknowledgments

We thank all administrators from all the study sites for their support.

## Author Contributions

**Conceptualization:** Angelina A. Joho, Frank Sandi, James J. Yahaya.

**Data curation:** Angelina A. Joho, Frank Sandi, James J. Yahaya.

**Formal analysis:** Angelina A. Joho, Frank Sandi, James J. Yahaya.

**Investigation:** Angelina A. Joho, Frank Sandi, James J. Yahaya.

**Methodology:** Angelina A. Joho, Frank Sandi, James J. Yahaya.

**Resources:** James J. Yahaya.

**Supervision:** Angelina A. Joho, James J. Yahaya.

**Validation:** Angelina A. Joho, James J. Yahaya.

**Visualization:** James J. Yahaya.

**Writing – original draft:** James J. Yahaya.

**Writing – review & editing:** Angelina A. Joho.

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
