## [Editor Report · Decision Letter 0]

26 Oct 2022

PGPH-D-22-01621

Determinants of Knowledge, Attitude, and Practice among Patients with Type 2 Diabetes Mellitus: a Multicenter Study in Tanzania

Dear Dr. Joho,

Thank you for submitting your manuscript to PLOS Global Public Health. After careful consideration, we feel that it has merit but does not fully meet PLOS Global Public Health’s publication criteria as it currently stands. Therefore, we invite you to submit a revised version of the manuscript that addresses the points raised during the review process.

We look forward to receiving your revised manuscript.

Kind regards,

Nasheeta Peer

Academic Editor

Journal Requirements:

1. In the online submission form you indicate that your data is not available for proprietary reasons and have provided a contact point for accessing this data. Please note that your current contact point is a co-author on this manuscript. According to our Data Policy, the contact point must not be an author on the manuscript and must be a third party. Please revise your data statement to a non-author institutional point of contact, such as a data access or ethics committee, and send this to us via return email. Please also include contact information for the third party organization, and please include the full citation of where the data can be found.

Additional Editor Comments (if provided):

Before proceeding to peer-review, please present/report the data by sex (men and women) with the related p-values, for all tables. The overall/total data should be retained as well.

Please provide the questionnaires used as appendices.
---

## [Decision Letter · Decision Letter 1]

15 Feb 2023

PGPH-D-22-01621R1

Determinants of Knowledge, Attitude, and Practice among Patients with Type 2 Diabetes Mellitus: a Multicenter Study in Tanzania

Dear Dr. Joho,

Thank you for submitting your manuscript to PLOS Global Public Health. After careful consideration, we feel that it has merit but does not fully meet PLOS Global Public Health’s publication criteria as it currently stands. Therefore, we invite you to submit a revised version of the manuscript that addresses the points raised during the review process.

We look forward to receiving your revised manuscript.

Kind regards,

Palash Chandra Banik, MPhil

Academic Editor

Journal Requirements:

Additional Editor Comments (if provided):

Reviewers' comments:

Reviewer's Responses to Questions

**Comments to the Author**

1. If the authors have adequately addressed your comments raised in a previous round of review and you feel that this manuscript is now acceptable for publication, you may indicate that here to bypass the “Comments to the Author” section, enter your conflict of interest statement in the “Confidential to Editor” section, and submit your "Accept" recommendation.

Reviewer #1: (No Response)

Reviewer #2: (No Response)

2. Does this manuscript meet PLOS Global Public Health’s publication criteria? Is the manuscript technically sound, and do the data support the conclusions? The manuscript must describe methodologically and ethically rigorous research with conclusions that are appropriately drawn based on the data presented.

Reviewer #1: Partly

Reviewer #2: Partly

3. Has the statistical analysis been performed appropriately and rigorously?

Reviewer #1: No

Reviewer #2: No

4. Have the authors made all data underlying the findings in their manuscript fully available (please refer to the Data Availability Statement at the start of the manuscript PDF file)?

Reviewer #1: Yes

Reviewer #2: No

5. Is the manuscript presented in an intelligible fashion and written in standard English?

Reviewer #1: Yes

Reviewer #2: No

6. Review Comments to the Author

Reviewer #1: 1. Line number is essential to make a comment.

2. Under the methodology section, study participants exclusion criteria the statement "Additionally, we excluded all patients who had prior knowledge on diabetes so as to make the study population homogeneous". How is this implemented? What types of knowledge measurements the authors used to assess? A written informed consent should be used even before the pilot study (prior knowledge assessment) despite 20 pilot study were conducted. In addition, the knowledge discrepancy is a useful determinant for policy makers with having the possible association. Hence, the authors should elaborate the statement as exclusion criteria is a key for study participants selection.

3. The study was conducted at eight study sites (three district hospitals and five regional referral hospital). I have a doubt for the selection methods of the study sites as the authors responded "because of here are no studies addressing knowledge, attitude and practice in patients with T2DM from the study areas in Tanzania." This implies there are data mentioning other health institution concerning the KAP assessment. Additionally, the population is not correctly sampled with an appropriate sampling techniques (formula). This is the heart of the study. If sampling frame is not appropriately stated, the study will not represent the population.

4. Under "Data collection method and tools" section, with referencing the Ethiopian research as a tool for measurements of KAP. However, the referenced Ethiopian article is not mentioning knowledge. The article was only assessed the attitude and practice with three section (Socio-demographic status, attitude, and practice) You can access the study from (https://bmcpublichealth.biomedcentral.com/articles/10.1186/s12889-020-08953-6#additional-information). Hence, the authors should use another knowledge measurement tools or update their reference.

Reviewer #2: Thank you to the authors for submitting their manuscript.

• Abstract needs to be revised as it can be more precise and irrelevant details can be removed

• In the abstract it is mentioned that level of knowledge is significantly associated with attitude and practice. However, no statistical results are shown and discussed in the manuscript

• “The reason for selecting these study sites is that there are no studies addressing knowledge, attitude and practice in patients with T2DM from the study areas in Tanzania” this line is repeated in the STUDY AREA section. Please revise this

• The study excluded the patients with prior knowledge, hence, selecting the sample of individuals with low level of knowledge. It will cause the magnitudes to be biased downwards resulting in biased estimates. Due to the randomization the chances of bias are minimized but excluding patients with knowledge will disrupt the randomization process

• Also, convince sampling is not the most suitable to obtain population representative sample

• In MATERIALS AND METHODS section, there is no mention of how the sample size of 979 calculated and through which methodology. Please provide the appropriate details in this section.

• In the discussion section needs revision. There are weak linkages between knowledge, attitude and practice in the manuscript.

• A lot of repetition in the discussion section. Please revise it.

• In the conclusion, there it is written that attitude is not influenced by knowledge, however, in the abstract it is written the knowledge it significantly associated with both attitude and practice. Also, again no statistical test or values are shown the paper regarding the association between knowledge, attitude and practice.

• Apart from chi square tests and why was no regression analysis performed for knowledge, attitude and practice with the socio-demographic variable collected?

• Overall, the writing of the paper needs significant improvements and the word count of the paper should be reduced. Lastly, there are a lot of repetition in the paper which needs to be adjusted.

7. PLOS authors have the option to publish the peer review history of their article (what does this mean?). If published, this will include your full peer review and any attached files.

**Do you want your identity to be public for this peer review?** For information about this choice, including consent withdrawal, please see our Privacy Policy.

Reviewer #1: No

Reviewer #2: No

---

## [Decision Letter · Decision Letter 2]

28 Apr 2023

PGPH-D-22-01621R2

Determinants of Knowledge, Attitude, and Practice among Patients with Type 2 Diabetes Mellitus: a Multicenter Study in Tanzania

Dear Dr. Joho,

Thank you for submitting your manuscript to PLOS Global Public Health. After careful consideration, we feel that it has merit but does not fully meet PLOS Global Public Health’s publication criteria as it currently stands. Therefore, we invite you to submit a revised version of the manuscript that addresses the points raised during the review process.

We look forward to receiving your revised manuscript.

Kind regards,

Palash Chandra Banik, MPhil

Academic Editor

Journal Requirements:

Additional Editor Comments (if provided):

Reviewers' comments:

Reviewer's Responses to Questions

**Comments to the Author**

1. If the authors have adequately addressed your comments raised in a previous round of review and you feel that this manuscript is now acceptable for publication, you may indicate that here to bypass the “Comments to the Author” section, enter your conflict of interest statement in the “Confidential to Editor” section, and submit your "Accept" recommendation.

Reviewer #3: (No Response)

Reviewer #4: All comments have been addressed

2. Does this manuscript meet PLOS Global Public Health’s publication criteria? Is the manuscript technically sound, and do the data support the conclusions? The manuscript must describe methodologically and ethically rigorous research with conclusions that are appropriately drawn based on the data presented.

Reviewer #3: Partly

Reviewer #4: No

3. Has the statistical analysis been performed appropriately and rigorously?

Reviewer #3: Yes

Reviewer #4: No

4. Have the authors made all data underlying the findings in their manuscript fully available (please refer to the Data Availability Statement at the start of the manuscript PDF file)?

Reviewer #3: Yes

Reviewer #4: Yes

5. Is the manuscript presented in an intelligible fashion and written in standard English?

Reviewer #3: No

Reviewer #4: No

6. Review Comments to the Author

Reviewer #3: Dear authors, I am reviewing the current version of the manuscript and have noticed that you have already made substantial revisions in response to comments from previous reviewers.

I have the following comments -

1) The language of the manuscript needs to be improved considerably. I will suggest that you ask someone with good command of English to do a proofreading and copyediting to improve the clarity of the language in the paper.

2) In the Introduction section, please revise "The International Federation of Diabetes (IDM)" to "The International Diabetes Federation (IDF)"

3) In the Introduction section, there is a sentence - "This tremendous and alarming drastic increase in the incidence of DM in developing countries in which Tanzania is included has been linked to substantial demographic change from traditional ways of living to Westernized and urbanized [3]". Please share the link for reference 3, as I would like to check whether the reference is appropriate or not.

4) Out of the 8 study sites, how many were public hospitals and how many were private?

5) How was the sample size of 979 calculated?

6) There are two sentences in the Results section which I could not understand - "Study participants who were working in private health facilities had significantly higher percentage of adequate knowledge (79.5%) than those working in the

health facilities owned by government (58.7%) (p<0.001)." and "The percentage of study participants with positive attitude working at referral health facility was significantly higher (53.2%) than that of study participants who were working at either district (50.3%) or regional health facility (44.8%) (p = 0.008) (Table 4)." Do you mean that the participants of the study were working at the health facilities? Are they not supposed to the patients who were visiting these health facilities for their treatment?

7) In the Discussion section, please explain why you decided to exclude all patients who reported to have been involved or participated in sessions for raising of awareness and knowledge on diabetes from the study, and what are the implications of this. Are there any other studies which have used a similar exclusion criteria? Is this to understand the knowledge, attitude and practices of the naive population? Is it to understand the kind of efforts that the Ministry of Health has to undertake in order to sensitize this vulnerable segment of the population and to assess what are the key messages to be incorporated in social and behaviour change communication (SBCC) campaigns for this population?

8) In the 3rd paragraph of the Discussion section, there is a sentence - "Furthermore, it has been found that patients who stay longer with T2DM have a significant high level of knowledge compared to patients with short duration of the disease". Please revise the language to "patients who were earlier diagnosed with diabetes" and "patients who were recently diagnosed with diabetes" or similar terminology

9) The first sentence in the Conclusion section is grammatically incorrect.

10) Reference 2 should be "International Diabetes Federation". Please correct the spelling of diabetes.

11) Please fix the style of Reference 15.

12) In the Abstract, there is a sentence which I found confusing - "Working at a district hospital was significantly associated with adequate knowledge (p<0.01), positive attitude (p<0.008), and appropriate practice (p<0.004)." This is similar to comment #6. Are the participants in your study the employees of the health facilities or the patients seeking care at these facilities? Since you have used "working at a district hospital", it implies that staff of a district hospital are more likely to have adequate knowledge, positive attitude and appropriate practice. But that does not make sense.

Reviewer #4: Thank you for the opportunity to review the manuscript by Joho, et al., who must be commended for their work on this hugely important topic.

1 . Is the manuscript presented in an intelligible fashion and written in standard English?

That science has to be communicated in the English language unfortunately disadvantages non-native speakers. Nonetheless, the manuscript’s standard of English could be much improved. Manuscript currently has grammatical errors, repetitions, and does not read easily.

2. Has the statistical analysis been performed appropriately and rigorously?

Authors state that they undertook both descriptive and inferential analyses. However, their results are largely, if not solely, descriptive. Analyses reported in Tables 3-5 do not warrant the interpretations or conclusions that are drawn/made in the manuscript.

For example, authors claim that “Positive attitude was increasing linearly with increasing level of education of study participants…” [p.41, Association of sociodemographic characteristics with attitude]. Chi-square tests (which were undertaken by authors) assess for overall differences in expected vs observed frequencies; they are not trend tests, which is what authors imply in much of their discussion of results.

The previous reviewer`s suggestion that authors use regression modeling for their inferential analyses was not heeded.

Results from inferential analyses are best reported with their corresponding 95% CIs.

Under “Sampling Method”, authors state that “…participants were consecutively recruited until the required total sample size of 979 was met.” However, sample size estimation is not reported.

Tables 2-5 in fact report the prevalence (or frequency) of “adequate knowledge”, “good attitude” and “appropriate practices”, respectively. Thus, these tables could be much simplified, and easier to understand, if only the “yes” outcome (of a dichotomous variable [yes/no]) were reported versus reporting both “yes” and “no” outcomes.

Reporting results of (unadjusted and adjusted) regression modeling in Tables 3-5 will be more informative than the descriptive summaries that are presently presented in these tables.

In summary, the authors` manuscript might be much improved if the assistance of a statistician were sought.

7. PLOS authors have the option to publish the peer review history of their article (what does this mean?). If published, this will include your full peer review and any attached files.

**Do you want your identity to be public for this peer review?** For information about this choice, including consent withdrawal, please see our Privacy Policy.

Reviewer #3: No

Reviewer #4: No

---

## [Decision Letter · Decision Letter 3]

14 Jul 2023

PGPH-D-22-01621R3

Determinants of Knowledge, Attitude, and Practice among Patients with Type 2 Diabetes Mellitus: a Multicenter Study in Tanzania

Dear Dr. Joho,

Thank you for submitting your manuscript to PLOS Global Public Health. After careful consideration, we feel that it has merit but does not fully meet PLOS Global Public Health’s publication criteria as it currently stands. Therefore, we invite you to submit a revised version of the manuscript that addresses the points raised during the review process.

Review comments to address:

The manuscript by Joho, *et al.,* refers. Overall, their results and conclusions represent an important contribution to the subject particularly as they come from an understudied and under-reported part of the world.

However, some revisions will be required and the below suggestions may be worth considering.

**Language**

While considerable progress has been made in improving the grammar, the language of the manuscript can still be considerably improved and made more concise.

One example is the simultaneous use of past and present tense *, “This study **aims to assess** the levels of knowledge…. This **was a **cross-sectional multicenter hospital-based study which **included**…” *[Abstract] . The assistance of someone with good command should be sought.

**Statistics/Data analysis **

Further improvements will be helpful in the reporting/presentation of the results. Examples include:

Sample size estimation – The Leslie Kish formula, as I understand it, is meant for calculating effective sample sizes in weighted surveys. Authors should  ensure they have correctly applied this.

Table 1 – Including mean (SD) or median (IQR) for age and duration of illness. This is stated in the “Data analysis” section but not reported. Also, frequency and percentage need not be two separate columns but one column “frequency (%)”.

Table 2 – These results can be better presented as a bar graph/plot with 95% CI spikes.

Tables 3, 4, 5 – It is customary to place the reference category (1.00) for any variable at the top, and not the bottom. Consistency in use of decimal numbers in reporting results will be helpful.

**Discussion**

The “Discussion” section can be made considerably shorter and more concise. In its current form, it largely reads as another literature review or “Background” section. A discussion of the implications of these results in the Tanzanian context will be more relevant and informative.

**References**

Reference #2, in the Introduction section, suggests that Tanzania`s T2DM prevalence increased from 5.7% to 12.3% in a space of **2 years**, from 2019 to 2021. This seems unlikely, an authors should double check this.

We look forward to receiving your revised manuscript.

Kind regards,

Elliot Koranteng Tannor, MBChB, FWACP, MPhil(Neph), Cert Neph(SA), MBA

Academic Editor

Journal Requirements:

2. Please ensure that the Title in your manuscript file and the Title provided in your online submission form are the same.

Additional Editor Comments:

Minor revision is required and do well to revert as soon as you can.

Reviewers' comments:

Reviewer's Responses to Questions

**Comments to the Author**

1. If the authors have adequately addressed your comments raised in a previous round of review and you feel that this manuscript is now acceptable for publication, you may indicate that here to bypass the “Comments to the Author” section, enter your conflict of interest statement in the “Confidential to Editor” section, and submit your "Accept" recommendation.

Reviewer #3: All comments have been addressed

Reviewer #4: All comments have been addressed

2. Does this manuscript meet PLOS Global Public Health’s publication criteria? Is the manuscript technically sound, and do the data support the conclusions? The manuscript must describe methodologically and ethically rigorous research with conclusions that are appropriately drawn based on the data presented.

Reviewer #3: Yes

Reviewer #4: Partly

3. Has the statistical analysis been performed appropriately and rigorously?

Reviewer #3: Yes

Reviewer #4: No

4. Have the authors made all data underlying the findings in their manuscript fully available (please refer to the Data Availability Statement at the start of the manuscript PDF file)?

Reviewer #3: Yes

Reviewer #4: Yes

5. Is the manuscript presented in an intelligible fashion and written in standard English?

Reviewer #3: Yes

Reviewer #4: Yes

6. Review Comments to the Author

Reviewer #3: All my comments have been addressed. I would like to appreciate the efforts of the authors in making multiple revisions and hereby recommend that the paper be accepted for publication.

Reviewer #4: Please see attached comments.

7. PLOS authors have the option to publish the peer review history of their article (what does this mean?). If published, this will include your full peer review and any attached files.

**Do you want your identity to be public for this peer review?** For information about this choice, including consent withdrawal, please see our Privacy Policy.

Reviewer #3: No

Reviewer #4: No

---

## [Editor Report · Decision Letter 4]

29 Aug 2023

PGPH-D-22-01621R4

Determinants of Knowledge, Attitude, and Practice among Patients with Type 2 Diabetes Mellitus: a Cross-sectional Multicenter Study in Tanzania

Dear Dr. Joho,

Thank you for submitting your manuscript to PLOS Global Public Health. After careful consideration, we feel that it has merit but does not fully meet PLOS Global Public Health’s publication criteria as it currently stands. Therefore, we invite you to submit a revised version of the manuscript that addresses the points raised during the review process.

You are required to work on the reviewers minor comments point-by-point and resubmit for the editors decision as soon as you can. 

We look forward to receiving your revised manuscript.

Kind regards,

Elliot Koranteng Tannor, MBChB, FWACP, MPhil(Neph), Cert Neph(SA), MBA

Academic Editor

Journal Requirements:

2. Please provide separate figure files in .tif or .eps format only and remove any figures embedded in your manuscript file. Please also ensure all files are under our size limit of 10MB.

Additional Editor Comments (if provided):

Reviewer 1 accepted your manuscript in its current form.

Reviewer 2  comments:

The manuscript by Joho, *et al.,* refers. Overall, their results and conclusions represent an important contribution to the subject particularly as they come from an understudied and under-reported part of the world.

However, some revisions will be required and the below suggestions may be worth considering.

**Language**

While considerable progress has been made in improving the grammar, the language of the manuscript can still be considerably improved and made more concise.

One example is the simultaneous use of past and present tense *, “This study **aims to assess** the levels of knowledge…. This **was a **cross-sectional multicenter hospital-based study which **included**…” *[Abstract] . The assistance of someone with good command should be sought.

**Statistics/Data analysis **

Further improvements will be helpful in the reporting/presentation of the results. Examples include:

Sample size estimation – The Leslie Kish formula, as I understand it, is meant for calculating effective sample sizes in weighted surveys. Authors should  ensure they have correctly applied this.

Table 1 – Including mean (SD) or median (IQR) for age and duration of illness. This is stated in the “Data analysis” section but not reported. Also, frequency and percentage need not be two separate columns but one column “frequency (%)”.

Table 2 – These results can be better presented as a bar graph/plot with 95% CI spikes.

Tables 3, 4, 5 – It is customary to place the reference category (1.00) for any variable at the top, and not the bottom. Consistency in use of decimal numbers in reporting results will be helpful.

**Discussion**

The “Discussion” section can be made considerably shorter and more concise. In its current form, it largely reads as another literature review or “Background” section. A discussion of the implications of these results in the Tanzanian context will be more relevant and informative.

**References**

Reference #2, in the Introduction section, suggests that Tanzania`s T2DM prevalence increased from 5.7% to 12.3% in a space of **2 years**, from 2019 to 2021. This seems unlikely, an authors should double check this.

---

## [Editor Report · Decision Letter 5]

2 Nov 2023

Determinants of Knowledge, Attitude, and Practice among Patients with Type 2 Diabetes Mellitus: a Cross-sectional Multicenter Study in Tanzania

PGPH-D-22-01621R5

Dear Ms. Joho,

We are pleased to inform you that your manuscript 'Determinants of Knowledge, Attitude, and Practice among Patients with Type 2 Diabetes Mellitus: a Cross-sectional Multicenter Study in Tanzania' has been provisionally accepted for publication in PLOS Global Public Health.

1. Do well to also attach an appropriately addressed cover letter as the attached cover letter in our records seem to be addressed to the International Journal of Diabetes in Developing countries. 

2. Also ensure all the tables have a legend with all abbreviations used. 

Best regards,

Elliot Koranteng Tannor, MBChB, FWACP, MPhil(Neph), Cert Neph(SA), MBA

Academic Editor
